# New Zealand River Hydrology under Late 21st Century Climate Change

**Daniel B. G. Collins** [1,2] 

[1]  National Institute of Water and Atmospheric Research, P.O. Box 8602, Christchurch 8440, New Zealand; daniel.collins@lincolnuni.ac.nz

[2]  Department of Environmental Management, Lincoln University, Lincoln 7647, New Zealand

**Abstract:** Climate change is increasingly affecting the water cycle and as freshwater plays a vital role in countries' societal and environmental well-being it is important to develop national assessments of potential climate change impacts. Focussing on New Zealand, a climate-hydrology model cascade is used to project hydrological impacts of late 21st century climate change at 43,862 river locations across the country for seven hydrological metrics. Mean annual and seasonal river flows validate well across the whole model cascade, and the mean annual floods to a lesser extent, while low flows exhibit a large positive bias. Model projections show large swathes of non-significant effects across the country due to interannual variability and climate model uncertainty. Where changes are significant, mean annual, autumn, and spring flows increase along the west and south and decrease in the north and east. The largest and most extensive increases occur during winter, while during summer decreasing flows outnumber increasing. The mean annual flood increases more in the south, while mean annual low flows show both increases and decreases. These hydrological changes are likely to have important long-term implications for New Zealand's societal, cultural, economic, and environmental well-being.

**Keywords:** climate change; river flow; national modelling; validation; New Zealand

## 1. Introduction

Climate change is increasingly affecting global, regional, and local water cycles, impacts that are projected to continue over the course of the 21st century [1–6]. How the water cycle responds to climate warming can vary significantly among regions and catchments due to differences in climatic and landscape characteristics and prevailing hydrological processes [7,8]. The resulting hydrological impacts can in turn present substantial implications for social, cultural, environmental, and economic well-being [1,9,10], prompting the need for mitigation and adaptation measures tailored to local and regional circumstances [11,12], while acknowledging impact uncertainties [13].

An important part of climate change adaptation is the use of impact studies to inform decision-making. For hydrological impacts, these studies often follow a similar top-down modelling cascade [14]: scenarios of future emissions, encapsulated by Representative Concentration Pathways (RCPs) [15], are used to drive General Circulation Models (GCMs), whose outputs are downscaled to a more useful resolution, bias-corrected, and then used to drive hydrological models (HMs). Studies vary in their choice of scale (global, continental, national, or catchment), and in their choice of RCPs, GCMs, downscaling, bias-correction, and HMs, each adding uncertainty to the final results [16]. While large-scale modelling or studies of a few illustrative catchments can provide broad results, for the studies to have direct relevance to decision-making they need to be conducted at commensurate resolutions and scales [17]. National catchment hydrological models are particularly useful in this regard [18] as they aim to capture the varying hydrological processes that distinguish rivers

from one another, painting a rich picture across hydrological gradients, catchment sizes, and flow recorder densities.

While national and trans-national hydrological impact studies have been conducted over many parts of the world [4,19–21], New Zealand is relatively understudied. New Zealand has warmed about 1 °C over the past century [22] and is projected to warm by another 0.1–4.6 °C by the end of this century [23]. Weak trends in precipitation have been detected in observational records, with increases on the west of the South Island and decreases on the east [24], while model projections point towards more precipitation in the west and south of the country, and less in the north and east [23]. National or semi-national hydrological impacts studies have been published in the grey literature [25–27], projecting a patchwork of increases and decreases in flows around the country, but the modelling lacked validation and their treatment of uncertainty was limited.

Advancing knowledge of climate change impacts on New Zealand river hydrology would have both domestic and international benefits. New Zealand's society, culture, environment, and economy are closely connected to river hydrology [28], and with the prospects of climate change, adapting to foreseeable hydrological effects are of national importance [29,30]. In addition, New Zealand's climate is relatively distinct from other nations in that it is dominated by a temperate maritime climate with distinct but muted seasonality, while also having significant alpine influences producing permanent snow and ice cover and extreme orographic effects [31]. Studying New Zealand's hydrological response to climate change would thus add to the general understanding of hydrological change.

The purpose of this study is thus to assess the potential impacts of 21st century climate change on New Zealand river flow regimes. Using a national climate-hydrology model cascade, natural river flows across 43,862 locations are simulated from 1971–2099 using four RCPs and six downscaled and bias-corrected GCMs. Components of the flow regime considered are mean annual and seasonal flows, mean annual flood (MAF), and mean annual 7-day low flow (MALF). Hydrological differences are assessed between the reference period (1986–2005) and late century (2080–2099). All projected effects are assessed for statistical significance in relation to combined GCM uncertainty and modelled inter-annual variability.

## 2. Materials and Methods

### 2.1. Study Region

The study area comprises the two main islands of New Zealand (Figure 1), the North Island and South Island, with an area of 264,000 km$^2$ and lying between 34° to 47° S in the Southwest Pacific; the country's remaining area of 8000 km$^2$ is spread across many much smaller islands with limited environmental data. Topography of the North Island is dominated by a central volcanic plateau that rises to 2797 m, and that of the South Island by the southwest/northeast-aligned Southern Alps that rise steeply on the west to a height of 3724 m [32]. Landcover of the North Island is predominantly pasture, with significant tracts of indigenous and exotic forest, while tussock, pasture, and indigenous forest dominate South Island vegetation [33].

The country is situated within the prevailing westerlies of the mid-latitude Southern Hemisphere [31]. Weather patterns are dominated by travelling anticyclones, depressions, and fronts within this flow, with extra-tropical cyclones having a rare but significant influence. Much of the country experiences a temperate maritime climate, while warm sub-tropical conditions occur in the north and both semi-arid and severe alpine climates inland in the south. Median annual temperatures vary between 2 °C, or lower, in the Southern Alps to 18 °C in the north [34]. Daytime summer temperatures typically range between 18 °C and 24 °C, while overnight winter temperatures typically drop to between −2 °C to 8 °C. Orographic effects combined with the prevailing westerly airflow produce high precipitation rates along the Southern Alps, in places over 10 m year$^{-1}$, with leeward areas dropping to 350 mm year$^{-1}$. The North Island and the north of the South Island have seasonal precipitation patterns with winter (June, July, August) maxima. Other parts of the South Island have

flatter seasonal cycles with either autumn maxima or bi-modal patterns (autumn/spring maxima and winter minima). Snow cover in the South Island varies from 5% in the summer to 35% in winter [35]. Interannual climate variability is high with a number of climate modes influencing the region, notably the El Niño-Southern Oscillation (ENSO), the Interdecadal Pacific Oscillation (IPO), and the Southern Annular Mode (SAM) [31,36].

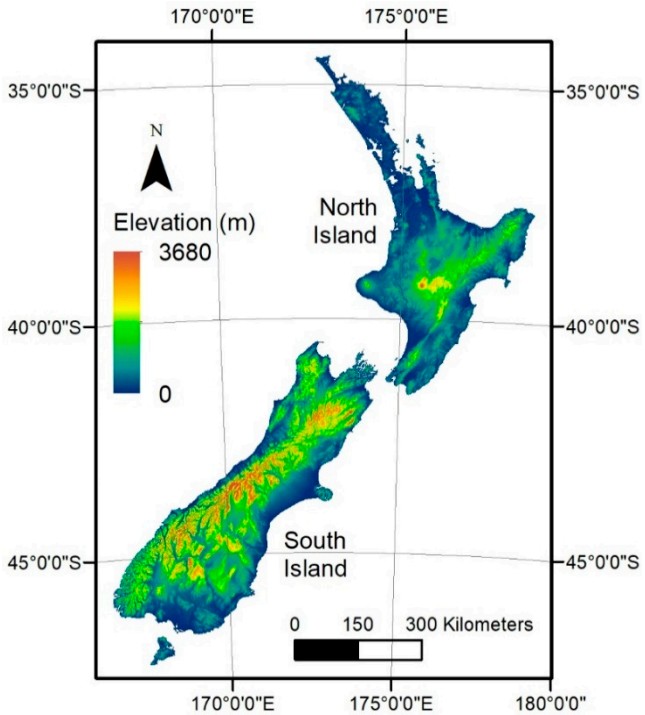

**Figure 1.** New Zealand location and topography (m).

Mean annual river flow to the coast accounts for 60–80% of the national water budget [37–39], while individual catchments span both energy- and water-limited conditions [38]. With the highest precipitation rates at higher elevations, alpine regions serve as water towers for New Zealand's major rivers. Seasonal snowmelt occurs over spring and summer [35] and contributes up to 17% of some rivers' mean annual flows at the outlets of snow-affected catchments [40], but is generally very small or non-existent. Monthly mean flows tend to peak in winter, moving through spring and into summer for southern alpine rivers, due primarily to the regional timing of precipitation but also to a lesser degree to springtime snowmelt [41]. Flow extremes vary with both climate and landscape. Peak flood flow rates, normalised by catchment area, are highest in rivers draining mountainous regions, particularly the Southern Alps, and lowest in leeward areas and those in the North Island underlain by rock of volcanic origin [42]. Times of concentration for rivers unaffected by lakes and storages are typically less than 12 h [43]. Low flows occur mainly in summer, while some alpine rivers of the Southern Alps tend to have winter minima [42]. Specific mean annual low flows are highest along the Southern Alps and across the volcanic region of the North Island; they are lowest along the east of both islands and the north of the North Island.

### 2.2. Climate Data

The climate data used in the hydrological modelling are described in Ministry for the Environment [23]. The data are derived from six General Circulation Models (GCMs) as part of the Coupled Model Intercomparison Project Phase 5 (CMIP5) [44], each driven by observed radiative forcings over the historical period and four Representative Concentration Pathways (RCPs) for the future [15]. The six GCMs (BCC-CSM1.1, CESM1-CAM5, GFDL-CM3, GISS-E2-R, HadGEM2-ES,

and NorESM1-M) were selected as they validated well on New Zealand's present climate while also being as different as possible from one another in the parent global model so as to span a likely range of climate sensitivity. The RCPs include a mitigation pathway (RCP2.6), two stabilisation pathways (RCP4.5 and RCP6.0), and a high-end pathway (RCP8.5). Relating these to the Paris Agreement thresholds of 1.5 °C and 2.0 °C [45], only RCP2.6 remains within the first threshold by the end of the century when averaged across all CMIP5 projections, and only RCP2.6 and RCP4.5 remain within the second [44].

Sea surface temperatures from the six GCMs were bias-corrected and used to drive the global atmosphere model HadAM3P [46] and then the 0.27° (~27 km) regional atmosphere model HadRM3P [47,48]. The regional output fields were then further downscaled, using seasonal quartile and statistical mapping, to an approximately 5 km grid at a daily time-step and bias-corrected relative to 1980–1999 climatology. Taking the data fields beyond the work presented by Ministry for the Environment [23], the daily 5 km precipitation were then adjusted following Woods, Hendrikx, Henderson, and Tait [38], which uses observed river flows to make corrections to the less certain precipitation values. Lastly, precipitation was stochastically disaggregated from daily to hourly, and daily temperature was disaggregated using a sinusoidal pattern [49].

### 2.3. Hydrological Modelling

The hydrological model used in this study is TopNet [18,49]—the only hydrological model currently parameterised and validated nationally for New Zealand. TopNet is a semi-distributed, process-based catchment model that simulates water storages and fluxes across the snowpack, plant canopy, rooting zone, shallow subsurface, lakes, and rivers at hourly time steps. Shallow groundwater returns to the river within the sub-catchment from which it originates; there are no deep or regional groundwater flows in the version of TopNet used for this analysis. Abstractions, diversions, return flows, impounds, and irrigation are not included, and thus the modelled river flows are considered 'natural'. The model is driven by time-series of precipitation, temperature, relative humidity, short wave solar radiation, mean sea level pressure, and wind speed.

Topography is derived using a 30 m Digital Elevation Model (DEM). Sub-catchment boundaries and river lines are obtained from the digital network of the River Environment Classification version one [50], with the smaller catchments aggregated up to Strahler 3 scale where possible. This results in 43,862 modelled river locations and corresponding contributing areas ranging from 900 $m^2$ to 122 $km^2$ and a mean area of 6 $km^2$. Landscape properties are parameterised based on the Land Cover Database version 2, the Land Resource Inventory, and the Fundamental Soil Layers [33] and are described in more detail in Clark, Rupp, Woods, Zheng, Ibbitt, Slater, Schmidt, and Uddstrom [49]. Sub-catchments are treated as homogenous averages of the landscape properties contained therein. Due to the paucity of some spatial information, some soil parameters are set uniformly across New Zealand [18].

Validation of TopNet based on the observed climate has been reported previously by Booker and Woods [51] and McMillan, Booker and Cattoen [18], based on slightly different implementations of the model. Mean annual flow and particularly MALF have shown positive biases, while MAF has shown both positive and negative biases depending on the inclusion of flow duration curve (FDC) based correction. McMillan, Booker and Cattoen [18] noted that seasonal patterns were reproduced well, and better model performance for larger and medium-wet catchments, and for those with smaller seasonal variations.

The simulations for the present study, using high-performance computing facilities, are split into two periods: 1971–2005 is driven by historical emissions forcing, and 2006–2099 is driven by the RCPs. The years 1986–2005 are used as the reference period against which the projections are compared. These are also the modelled years used in the validation. To illustrate the potential climate change effects in this study, the late-century period from 2080–2099 is used.

Seven hydrological indicators are calculated from the hourly model output over 20-year periods: mean (annual) flow ($\overline{Q}$), mean summer, autumn, winter and spring flows ($\overline{Q}_{Sum}$, $\overline{Q}_{Aut}$, $\overline{Q}_{Win}$, $\overline{Q}_{Spr}$),

mean annual flood (MAF), and mean annual 7-day low flow (MALF). The calendar year is adopted for MAF and MALF, and MAF is derived from hourly simulated data to be as comparable to instantaneous observations as possible.

*2.4. Validation*

Confidence in modelled impacts relies on how the models perform when evaluated during the observational period [52]. It is standard for models within a climate-hydrology cascade to be evaluated individually, in order to ensure the models provide useful reflections of the climate or hydrology of the study area, but there is also benefit in evaluating the whole model cascade from start to finish [14]. Validating the whole model cascade over reference period simulations provides a truer measure of model performance against which projections of climate change impacts are to be compared. A challenge when modelling national climate change impacts, however, is scaling the evaluation method up from several catchments to hundreds or thousands of catchments. A potential solution to this may be adopted from the validation of national hydrological models under observed climate drivers, such as by McMillan, Booker, and Cattoen [18] who examined between-site differences in hydrological signatures.

Validation of sub-models within the climate-hydrology model cascade considered here has already been carried out separately. The Ministry for the Environment [23] evaluated 41 GCMs and identified six that validate well on New Zealand's observed climate. The downscaling and bias-correction were validated against a suite of land- and satellite-based observations by Ackerley, Dean, Sood, and Mullan [48]. Booker and Woods [51] and McMillan, Booker and Cattoen [18] evaluated the HM's ability to reproduce observed river flow characteristics based on observed weather. Given that the GCM simulations are free-running, however, modelled events during the historical period do not correspond to observed events, and so time-series cannot be directly validated against one another as is common for hydrological models. Nor can multi-year dry or wet periods be compared. Validation of the whole model cascade must thus compare observed and modelled river flows nationally at more climatological time-scales using hydrological metrics.

Sites used for validation are selected from a network of over 2000 across the country. Sites are excluded if they are significantly influenced by abstractions, diversions, return flows, or impounds, if the nature of the recorder operation and rating curves are not suited to the hydrological indicator in question, if they do not directly correspond to river reaches in the largely Strahler 3-based modelled digital network, or if the digital and real upstream catchment areas differ by more than 5%. For each usable site, data are then processed into annual series. For the mean flows a particular year is included only if it has at least 95% data coverage. For MAF and MALF, a year is only included if, by visual inspection, any data gaps are unlikely to have included the true peak or low flows. For MAF, this inspection is aided by comparing the data with up to three of the closest recorder sites with similar catchment areas. Furthermore, observed data are only considered prior to 2006, to correspond to the transition between 'historical' and 'future' climate change simulations, and if sites have at least 10 years of usable data. This leaves 338 sites for $\overline{Q}$, 313, 312, 311, and 312 sites for $\overline{Q}_{Sum}$, $\overline{Q}_{Aut}$, $\overline{Q}_{Win}$, and $\overline{Q}_{Spr}$, respectively, and 601 and 280 sites for MAF and MALF, respectively. All of these are compared to modelled results from 1986–2005, the reference period against which the climate change projections are compared. Validation is assessed using the percentage bias (PBIAS) and Nash-Sutcliff Efficiency (NSE) for the mean hydrological metrics across the recorder sites. It should be noted that values of these metrics cannot be compared directly with those obtained from at-site time-series validation.

*2.5. Climate Change Impact Analysis*

The effects of climate change on the hydrological metrics are inferred by comparing the reference period to the future period, both of which are 20 years long. Effects are reported as multi-model (GCM) mean percentage changes. Effect significance is assessed using the two-sided *t*-test [53] with a significance level of 5% applied to the pooled multi-model and multi-year data, which serves to isolate

potential climate change effects from the combined effects of modelled climate variability and climate model uncertainty. In using the multi-model mean, no presumption is made about which GCM is more reliable than any other in making climate change projections.

## 3. Results

### *3.1. Validation*

Comparing the modelled statistics against the observed (Figure 2) we see a range of model performances depending on the metric. For the mean annual and seasonal flows there is little scatter about the 1:1 line, with NSE ranging from 0.89 to 0.97 (Table 1). As this is applied nationally, this suggests a good ability to distinguish catchments from one another. PBIAS varies from the lowest (absolute value) of −7% for summer to the highest of 48% for autumn. Examining the spatial structure in the bias (Figure 3), there is a tendency for western, higher-precipitation areas to underpredict flows, while eastern and dryer areas as well as north-central North Island tend to overpredict. The effect of this is to reduce the modelled spatial variability of mean river flows.

The largest scatter corresponds to MAF, indicating the lowest performance among the seven metrics at distinguishing catchments from one another, and implying the lowest confidence in the spatial detail of the projections. Examining the distribution of biases across the country, there is a marked overprediction in north-central North Island in an area of volcanic geology and soils. TopNet is known to perform less well for floods in this region [18]. Nationally, however, the PBIAS of −21% and NSE of 0.91 are still reasonable. Looking closer at the scatter plot shows that smaller floods (smaller catchments) tend to have a positive bias, while the mid-to-large floods are negatively biased.

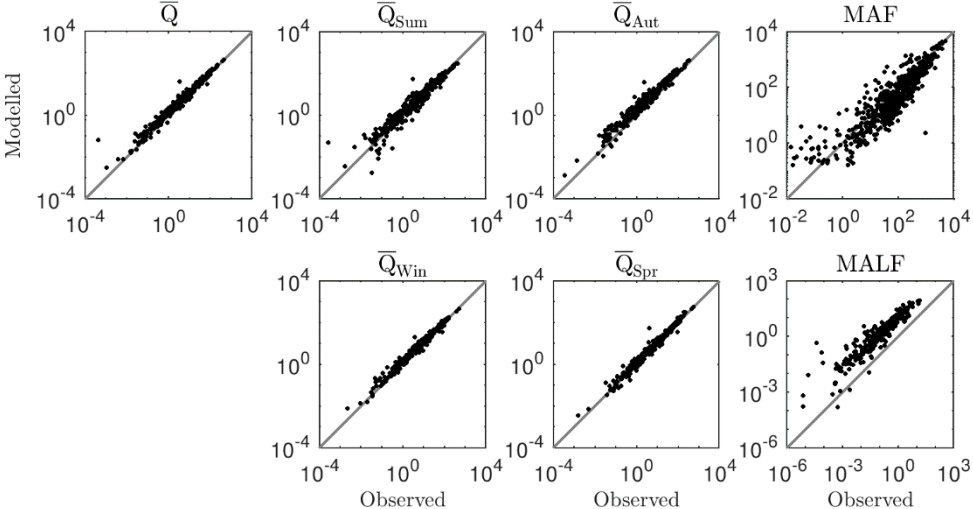

**Figure 2.** Scatter plot of modelled vs. observed values of the seven flow descriptors: mean flow ($\overline{Q}$), mean summer, autumn, winter, and spring flows ($\overline{Q}_{Sum}$, $\overline{Q}_{Aut}$, $\overline{Q}_{Win}$, $\overline{Q}_{Spr}$), Mean Annual Flood (MAF), and Mean Annual 7-day Low Flow (MALF).

**Table 1.** National PBIAS and NSE for the seven flow descriptors.

| Metric | PBIAS (%) | NSE |
|:---:|:---:|:---:|
| $\overline{Q}$ | 19% | 0.92 |
| $\overline{Q}_{Sum}$ | −7% | 0.89 |
| $\overline{Q}_{Aut}$ | 48% | 0.96 |
| $\overline{Q}_{Win}$ | 12% | 0.97 |
| $\overline{Q}_{Spr}$ | 23% | 0.95 |
| MAF | −21% | 0.91 |
| MALF | 862% | 0.60 |

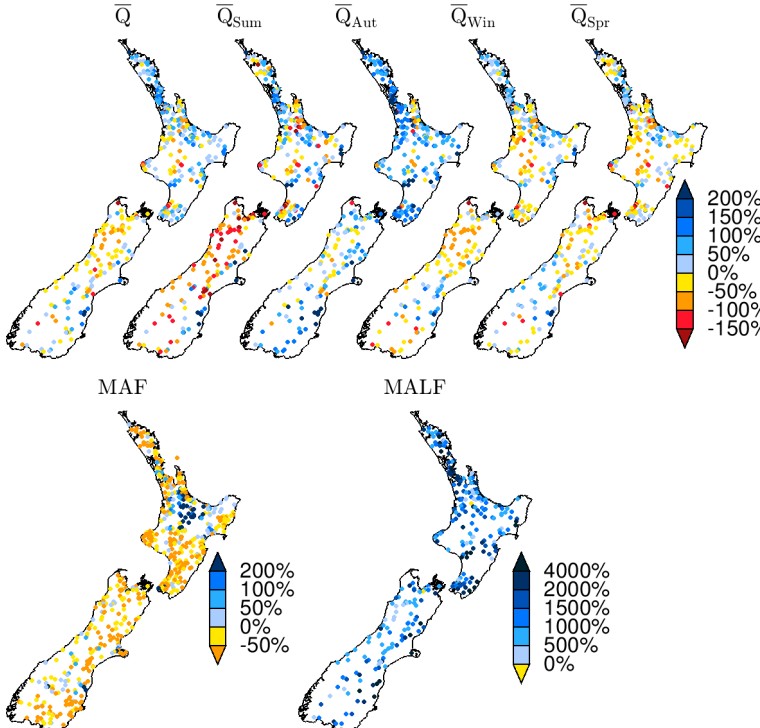

**Figure 3.** At-site percentage bias (PBIAS) for the seven flow descriptors: $\overline{Q}$, $\overline{Q}_{Sum}$, $\overline{Q}_{Aut}$, $\overline{Q}_{Win}$, $\overline{Q}_{Spr}$, MAF, and MALF.

The largest PBIAS (862%) and lowest NSE (0.60) stem from MALF, indicating that the model cascade does a very poor job reproducing low flows. This poorer performance has been noted before for TopNet [18] and for other models used for climate change impact studies [52], and is likely more a reflection of the HM than the climate input. This high national bias is offset, in a way, by the consistency of the bias among catchments as seen in the roughly vertical shift in the observed-modelled scatter plot (Figure 2), echoed in the not unreasonable NSE, which indicates that the bias is relatively systematic across the country. Thus, while the absolute values of modelled MALF are not realistic, calculating relative changes in MALF would in effect cancel the systematic bias out. Turning to the spatial structure in the biases, those areas that exhibit overpredictions for the mean annual and seasonal flows tend to produce higher overpredictions in MALF.

Taking these validation results together, the climate-hydrology model cascade produces reasonable reflections of observed patterns for most metrics considered here, particularly considering the hydrology model being uncalibrated, and are consistent with previous validation studies. MALF is an obvious exception in terms of absolute values; however, it may still be used with care to assess relative differences due to climate changes. Caution is advised if any future studies using these hydrological projections were to consider metrics that are derived from MALF. Some caution is also advised in interpreting effects among regions considering the spatial structure in the biases nationally.

*3.2. Changes in Mean Annual and Seasonal Flows*

Turning now to late-century projections of hydrological change, we see that changes in mean annual flow generally show increases across the west and south of the South Island with some decreases in the north and east of the North Island (Figure 4 and Table 2). The South Island's increases become larger and more extensive increases under higher emissions scenarios, in places exceeding +20%, while the North Island increases in the west give way to decreases in the north and east of the island. There is also a band of non-significant differences that run southwest-northeast to the east of the South

Island mountain range under all RCPs. The linear river extent of significant differences, as a fraction of total national river length, increases from 25% of the country under RCP2.6 to 68% under RCP8.5.

**Table 2.** Percentage of the country's total river length that exhibits significant changes. Numbers before commas correspond to increases and after commas to decreases.

| Metric | RCP2.6 | RCP4.5 | RCP6.0 | RCP8.5 |
|---|---|---|---|---|
| $\overline{Q}$ | 25, 0 | 30, 10 | 46, 10 | 52, 16 |
| $\overline{Q}_{Sum}$ | 2, 5 | 2, 11 | 6, 9 | 9, 24 |
| $\overline{Q}_{Aut}$ | 2, 0 | 12, 1 | 22, 2 | 28, 8 |
| $\overline{Q}_{Win}$ | 37, 0 | 46, 5 | 61, 2 | 66, 11 |
| $\overline{Q}_{Spr}$ | 10, 0 | 16, 16 | 32, 20 | 37, 18 |
| MAF | 10, 0 | 17, 0 | 37, 0 | 58, 0 |
| MALF | 9, 5 | 14, 18 | 16, 22 | 16, 47 |

Changes in mean summer flow show reductions in parts of the North Island and inland parts of the South Island, and decreases along the west and east coasts of the South Island (Figure 4). This pattern is accentuated under RCP8.5, with changes exceeding both +20% and −20%, but is essentially absent under RCP2.6 due to the extensive coverage of non-significant differences. Significant differences under RCP8.5 cover 33% of the national stream length, although in general summer sees the least extensive significant differences of all the seasons (Table 2).

Changes in mean autumn flow resemble a muted version of the mean annual flow changes (Figure 4 and Table 2). By late-century, only 2% of the country exhibits increases under RCP2.6, with no decreases, while increases along the west, south, and east of the South Island develop from RCP4.5 to RCP8.5 (28% of the country). Decreases, mostly in the North Island, expand to cover only 8% under RCP8.5. Again, there is a band running southwest-northeast to the east of the South Island's main divide that shows no change.

Winter sees the greatest changes in mean flow of all the seasons, in both magnitude and extent of significant changes (Figure 4 and Table 2). Only during winter do statistically significant changes in mean seasonal flows exceed 50% of the country—by late-century under RCP6.0 and RCP8.5. Pronounced increases run along much of the west of the South Island, increasing in magnitude from RCP2.6 to RCP8.5 with some changes over +50% and enveloping most of the island. The larger east-flowing, alpine-fed rivers in the South Island also show increases, despite passing through areas with no significant flow changes. Western-central parts of the North Island also exhibit increases, albeit smaller and only emerging extensively under RCP6.0 and RCP8.5. Decreases in the north and east of the North Island, some greater than −10%, essentially only emerge under RCP8.5.

Changes in mean spring flow are intermediate between autumn and annual changes (Figure 4 and Table 2). Some North Island increases are projected under RCP2.6, but they diminish under higher emissions scenarios while decreases expand in the north and east of the country. In the South Island, there is no substantial change under RCP2.6, while increases in flow develop in the west and south of the island under RCP4.5, becoming more extreme and extensive towards RCP8.5. There are also some decreases in flow in the northeast of the island, inland.

Comparing these results to the validation maps (Figure 3), we see that areas where mean flows are projected to increase tend to occur where the modelled reference period produces underestimates. Similarly, projected decreases occur where the modelled reference period is overestimated. This implies that the regional differences between increases and decreases may not be as stark as is modelled.

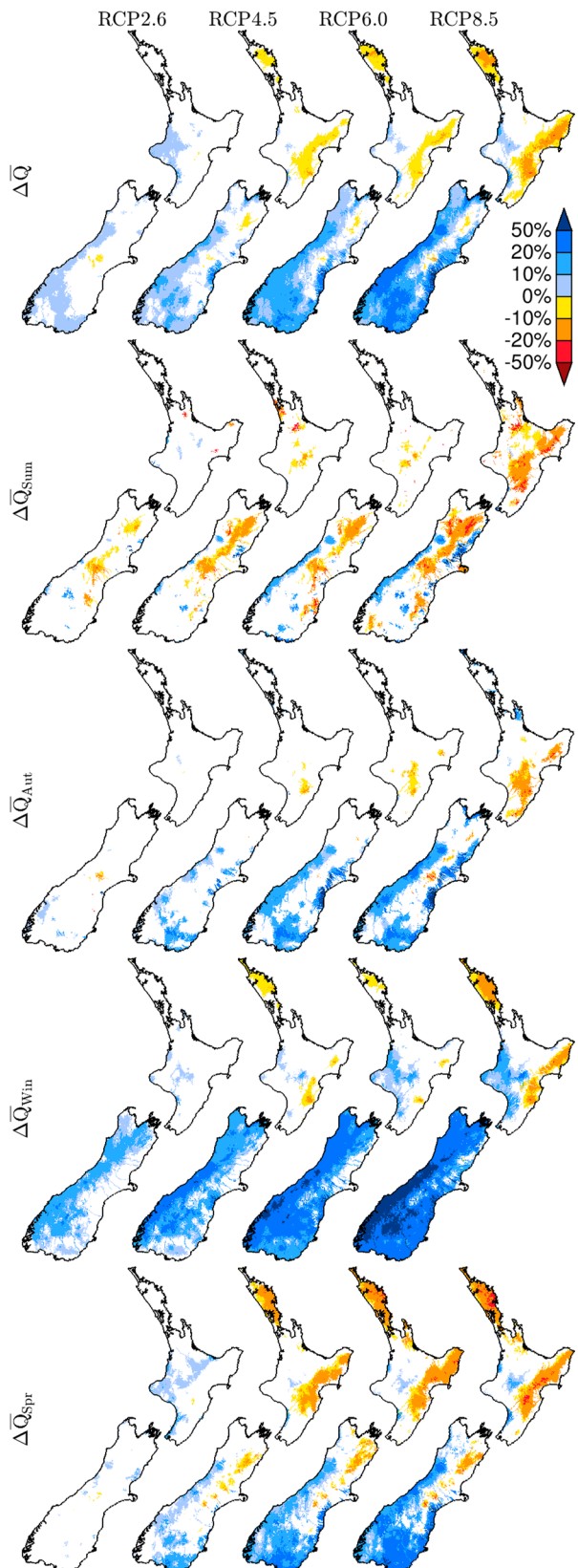

**Figure 4.** Late-century percentage changes in $\overline{Q}$, $\overline{Q}_{Sum}$, $\overline{Q}_{Aut}$, $\overline{Q}_{Win}$, and $\overline{Q}_{Spr}$ across four Representative Concentration Pathways (RCPs) relative to the reference period. White indicates statistically insignificant differences.

### 3.3. Changes in High and Low Flow Extremes

Projected changes in MAF show late-century increases becoming more pronounced and extensive moving from RCP2.6 (10% extent) to RCP8.5 (58% extent) (Figure 5 and Table 2). Changes are concentrated in the south and then west of the South Island, with some areas increasing by over 50%, with little change in the North Island. Under all RCPs there are some larger rivers with significant changes that stand out conspicuously from non-significant neighbours, illustrating that effects may start and propagate downstream from alpine sources. While there are small pockets of decreases in MAF, they are too small and isolated to be compelling. It is noteworthy that the pocket of positive biases located in the centre-north of the North Island (Figure 3) shows little sign of change one way or another.

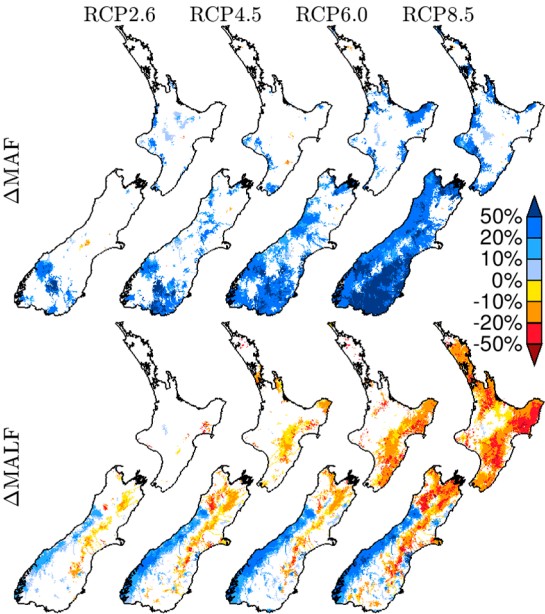

**Figure 5.** Late-century percentage changes in MAF and MALF across four RCPs relative to the reference period. White indicates statistically insignificant differences.

For MALF a coherent pattern emerges of increases and decreases, ramping up from RCP2.6 to RCP8.5 (Figure 5 and Table 2). Increases are projected for the west of the South Island, largely on the western flanks of the Southern Alps. Decreases are projected along an inland swathe to the east of the main divide and in the north of the South Island, as well as eastern, northern, and southern parts of the North Island. Changes more extreme than +50% and −50% are present, and the linear river extent of significant changes by late century under RCP8.5 rises to 16% for increases and 47% for decreases. The pattern roughly resembles the projected changes in mean summer flow but is more pronounced and more extensive. Moreover, while MALF exhibits a large positive bias (Figure 2), the relative uniformity of the bias across the country means that the patterns and relative scales of projected change are more likely to be believable.

## 4. Discussion

### 4.1. Interpreting Hydrological Changes

The projected hydrological changes are largely consistent with previous New Zealand studies, limited though they are. The results imply the shift in seasonality of lower flows from winter towards summer, as well as the overall increase in mean flows, for the Southern Alps-fed, southeast-flowing Clutha River/Mata-Au identified by Poyck et al. [54] and Jobst et al. [55], and show the shift occurring over a larger area of the South Island. Higher mean annual flows in the west and south of the South

Island, and lower in the north and east of the North Island, echo the work of Collins [25], although a notable difference is that lowland river flows along the east of the South Island are here projected to increase instead of decrease. The more complete national coverage and more in-depth analysis also provide a richer set of projections than was previously produced by Collins and Zammit [27] and Collins, Montgomery, and Zammit [26], importantly including the delineation of statistically non-significant effects.

Comparing the results to international studies is hampered by the uniqueness of regional circulation patterns, although several points of similarity are worth noting. There is sufficient uncertainty in many locations to prevent identification of a direction of change, let alone magnitude, although this uncertainty shrinks under more severe warming [21]. Of the changes that are significant, most of them occur in alpine or snow-affected catchments [8,56,57], but not necessarily both. This is partly due to increases in temperature shifting precipitation from snow to rain and melting snow earlier, and partly due to orographic effects on precipitation, which play a large role in New Zealand. The increases in MAF are also a relatively common projection under climate change, as warmer air can hold—and subsequently precipitate—more moisture [56].

The projections of hydrological change also align with the climatic changes reported for New Zealand by Ministry for the Environment [23], as is expected given the use of common GCM and RCM output. Changes in mean seasonal flows generally coincide with changes in precipitation but are modulated by increases in temperature. Higher increases in temperature tend to suppress the influence of higher precipitation, and greater changes in mean seasonal flows result from reductions in precipitation rather than from equivalent increases. This stems from the importance of evaporation in catchment water balance, and echoes analysis by Berghuijs et al. [58]. Changes in MAF bear a strong resemblance to changes in the 99th percentile of daily precipitation, as would be expected given that MAF is about the 99th percentile of the annual flows or rarer and that times of concentration are typically shorter than a day. However, there are locations where large increases in one are not matched by similarly large increases in the other. This suggests the importance of landscape features in the generation and propagation of floods, such as snowmelt and antecedent soil moisture conditions, beyond simply extreme precipitation. Changes in MALF do not strongly resemble any one of the climatic changes reported by the Ministry for the Environment [23] but rather more a complex combination of the annual number of dry days (with precipitation below 1 mm/day), the potential evapotranspiration deficit (PED; accumulated over the July–June water year), and the mean dry season precipitation, with the dominant climate metric differing across the country. Analysing the sensitivity of the hydrological changes to different driving factors (e.g., global or national mean surface temperature change, or mean sea level pressure change) would aid in understanding both the robustness of the results and the causal chain leading to them.

Some part of a causal chain may be gleaned from Ministry for the Environment [23]. During winter, Mean Sea Level Pressure (MSLP) is generally projected to decrease, particularly over the south of the South Island, resulting in stronger moisture-laden westerlies over the middle of the country. During summer, MSLP is projected to increase, particularly to the south-east of the country, which favours more north-easterly airflow and anticyclonic (high pressure) systems. Patterns of MSLP change during autumn and spring are less consistent but tend to resemble those of summer and winter, respectively. This helps to explain the seasonal patterns among the hydrological changes, which are modulated, as discussed above, by topography.

## 4.2. Uncertainties and Limitations

In conjunction with assessments of hydrological impacts, it is also vital to consider uncertainties and limitations. The validation results provide a succinct assessment of total model performance, in this case showing fair reproduction of reference period mean flows and MAF, and a large systematic bias in MALF. Validation in this way requires the modelling to be conducted at a sufficiently broad scale to encapsulate many gauging locations, catchment sizes, and hydrological conditions, and so is

not an option for studies of one or a few isolated catchments. Using these validation results to interpret the model projections in turn allows us to refine our conclusions considering overall model cascade uncertainty. This end-to-end validation can sit alongside other methods to better understand projection credibility [52]. However, validating on historical means does not provide as much confidence as validating also on historical fluctuations [52]. This is constrained by the shortness of the historical simulations but should be addressed in future climate change impact studies.

While assessing the full model cascade uncertainty is a valuable addition to climate change impact studies, it is not meant to replace analysis of uncertainties and limitations of the cascade's consistent steps, however, but to complement them. Climate change impact assessments are subject to uncertainties along the modelling cascade from the scenarios, GCMs, and downscaling, to the HMs. Globally, Hagemann et al. [59] showed that the spread of projected runoff changes was dominated variously by the scenario, GCM, and HM, depending on the location; for New Zealand's North Island it was largely the HMs, and for the South Island it was the GCMs. Across 12 catchments studied by Vetter et al. [60], the greatest source of uncertainty for three hydrological metrics was from GCMs, followed by RCPs, and lastly HMs, although the relative contributions varied among the catchments. For New Zealand's largest catchment, Jobst, Kingston, Cullen, and Schmid [55] showed that the GCM was the primary factor, followed by scenario, bias correction, and lastly the snow model. Which source dominates the uncertainty depends on the local hydrological processes and climatic changes [61,62].

Choosing all four RCPs [15] allows us to account for scenario uncertainty relatively well compared with other hydrological impacts studies, many of which use just one or two scenarios. Projections using RCP2.6 show that there may be significant hydrological impacts of climate change in some parts of New Zealand, even with an optimistic 1 °C of additional warming by the century's end. Under RCP4.5, results show that missing the 1.5 °C Paris Agreement target but achieving the 2 °C target may produce substantially more hydrological change, particularly in mean spring flows and MALF. RCP6.0 tends to lead to even more hydrological change. In addition, RCP8.5 yields striking changes in both extent and magnitude across the metrics considered, leading to a very different New Zealand, hydrologically. It should be remembered, however, that these RCPs are just four hypothetical futures of an effectively infinite set and that an alternative to using RCPs would be to partition climate projections by their mean global temperature changes above industrial levels [4].

Due to the criteria in selecting the six GCMs, climate model uncertainty should be reasonably well accommodated in these results. Any regional or local study will be constrained by the number of GCMs that validate satisfactorily in the domain of interest, as these six do. A promising development on the horizon is the work by Williams et al. [63] in building an earth systems model tailored to New Zealand's regional climate. However, two particular shortcomings of the GCMs in general remain a concern, namely uncertainties in simulating precipitation [64] and the poor or absent representation of climate modes and extratropical cyclones [23].

In contrast with the global climate modelling, the downscaling and bias-correction steps used in the driving climate data did not use alternative models or methods. The choice of downscaling can have an impact on hydrological projections [65], particularly if the method does not account for orographic effects [66], and the use of just a single climate model in the dynamic downscaling here is a particular weakness of the climate projections [23,67]. Similarly, the global study by Iizumi, et al. [68], comparing two bias-correction methods, showed that while the choice of methods contributed little to the uncertainties in temperature projections, the choice of methods was a major contributor to precipitation uncertainties.

In light of Hagemann, Chen, Clark, Folwell, Gosling, Haddeland, Hanasaki, Heinke, Ludwig, Voss and Wiltshire [59] and others, the use of just a single hydrological model is a notable shortcoming of the present study, although as yet only one physically based hydrological model has been parameterised across New Zealand. The late-century mean flow results under RCP4.5 do resemble the empirical, middle-of-the-road scenario results of Collins [25]. Moreover, while the model here does validate reasonably well on historical observations of mean flows and MAF, it exhibits a large systematic

bias in MALF. A range of parameterisations and/or process models, as well as improvements to the evaporation and slow pathway processes [18], may be advisable in order to better constrain projection uncertainties in future impact studies. Local calibration of the model where possible may also improve accuracy and stakeholder confidence, although Mendoza et al. [69] found that calibration does not necessarily improve agreement among HMs.

Looking across the model cascade from GCM to HM, the use of a single realisation per RCP-GCM pair means that the distribution of hydrological effects due to internal climate variability may be poorly sampled [70]. This would be most problematic for low-frequency events such as the 1% Annual Exceedance Probability (AEP) flood. For the hydrological means estimated here, however, a 20-year sampling period should be acceptable for an approximately quasi-steady-state condition, although 30 years or more would be preferred [71]. That said, even 30 years would be too short to properly sample climate modes such as IPO [72].

In terms of spatial resolution, the hydrological model produces results at an average resolution of 6 km$^2$. While this precision is useful for implementing the model's hydrological processes and is of interest to local stakeholders, it should not be conflated with accuracy. Datasets that underlie the HM are generally described at coarser resolutions, and there is spatial structure in the biases. The validation scatter for MAF, in particular, implies a degree of spatial imprecision in the simulation of floods. What resolution the results should be reported at, in order to provide useful information without overstating the level of knowledge, is unclear at this stage and warrants further research.

Finally, it is acknowledged that a formal quantification of uncertainty has not been performed here. This would help to better constrain the robustness of the model results [73]. However, as only one method or model is currently available each for the down-scaling, bias-correction, and hydrological modelling steps, thorough quantification of uncertainty is not possible at this stage.

*4.3. National Coverage*

The national extent of this study offers several advantages over typically much coarser impact studies. Firstly, it allows a high proportion of gauging locations to be used in the validation, which helps to constrain climate-hydrology model cascade uncertainty. Secondly, it yields an array of rich mapped results with enough detail to see different effects between neighbouring rivers, large and small. This can shed valuable light on the effect of catchment size on sensitivity to climate change. Thirdly, it can provide predictions that may be tested across the gauging network. This information could be used to manage a network of climate change monitoring sites, to constrain climate change detection analysis, or to test model performance. Fourthly, it can offer detailed information at both gauged and ungauged locations, which allows the impacts and implications of climate change to be addressed in a more detailed and location-specific manner. All of this is contingent, however, on the availability of sufficiently detailed spatial data and of high-performance computing facilities.

## 5. Conclusions

This research presents the validation of a climate-hydrology model cascade and its application in projecting changes in 21st century river hydrology across New Zealand. While the cascade itself comprises models with varying uncertainties, validation of the whole cascade using national hydrological modelling and long-term mean hydrological indicators allows the results to be evaluated succinctly. Comparing long-term mean indicators—observed and modelled—shows mean annual flows, mean seasonal flows, and MAF to be reproduced well, while MALF exhibits a large systematic bias.

The use of a national model also offers a powerful means of projecting climate change impacts for ungauged catchments. Comparing differences between the reference period of 1986–2005 to the late-century (2080–2099) projection period, we see that results show increases in mean annual river flow along the west and south of the South Island, in places exceeding 20%, and decreases exceeding 10% in the north and east of the North Island, with statistically significant changes becoming larger and more extensive under higher RCPs. Winter exhibits the largest and most extensive significant increases,

in many places over 50% under RCP8.5, due to the strong signal of increasing moisture-bearing westerly airflow across the South Island, while weakening easterlies deprive the north and east of moisture. The significant changes during summer are dominated by decreases in mean flow, while spring and autumn changes tend to reflect the regional patterns expressed in mean annual flow. Significant changes in MAF are almost all positive, lying mostly in the south and west of the South Island, with some increases exceeding 50%. The few statistically significant decreases are too isolated and restricted to be robust considering model uncertainties. MALF increases along the west of the South Island and decreases in eastern and northern parts of the North Island, and northern and inland parts of the South Island, again with changes becoming more accentuated late-century and under higher RCPs. The increased MALF along the south-west of the South Island is a consequence of substantial increases in dry season (winter) flows. In general, non-significant differences tend to outnumber significant differences across the hydrological indicators and RCPs; only for mean winter flows and MAF do statistically significant differences cover more than half of the country, and only for the higher emissions scenarios.

Uncertainties in the projections come from a range of sources, but perhaps the greatest source of unidentified uncertainty comes from the use of just a single hydrological model. It is thus recommended that alternative models and/or alternative process representations and parameterisations be used to assess the robustness of the climate change impacts to our understanding of New Zealand hydrological processes. There is also a concern regarding the spatial resolution of the results. While high-resolution modelling offers detail at the scale of much water resource and hazard decision-making, such precision may offer false confidence. It is thus also recommended that future research identify the minimum spatial scale at which the results are meaningful and valid. In the interests of effective communication to wide audiences, it may also be advisable to identify an equivalent maximum spatial scale before averaging of effects starts to hide important differences.

Lastly, in light of the significance of water resources and weather-related hazards to New Zealand [74] and of the magnitude of changes projected in this study, it is recommended that implications of these hydrological changes be carried further down the impacts cascade. This would include water resource availability, irrigation demand, hydropower generation, flood hazards, ecological and economic disruption, and ultimately adaptation. This should include times of emergence of significant impacts for different parts of the country and for different aspects of the water cycle, which would have implications for the timing of the deliberation and implementation of adaptation options.

**Funding:** This research was funded by the Deep South National Science Challenge under MBIE contract C01X1412.

**Acknowledgments:** Thanks to Christian Zammit for the original hydrological simulations, carried out under 'Climate Changes, Impacts and Implication' (MBIE contract C01X1225), and to Roddy Henderson, Christian Zammit, and four anonymous reviewers for helpful feedback on the manuscript.

**Conflicts of Interest:** The author declares no conflict of interest. The funders had no role in the design of the study; in the collection, analyses, or interpretation of data; in the writing of the manuscript; or in the decision to publish the results.

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
