# Peer review of "New Zealand River Hydrology under Late 21st Century Climate Change"

_water, doi:10.3390/w12082175_

Round 1
Reviewer 1 Report
Peer review of New Zealand river hydrology under late 21st century climate change
by Daniel B. G. Collins
Summary
The potential effects of climate change on freshwater in New Zealand were studied using a climate-hydrology model cascade at high spatial and temporal resolution. The TopNet hydrologic model, previously parameterized throughout New Zealand, was driven by downscaled and bias corrected meteorological data from six CMIP5 global climate models, each representing four RCP scenarios. Simulations spanned 1971 to 2099. Climate change impact was assessed by comparing hydrologic indicators calculated during the validation period (1986-2005) to indicators calculated at the end of the century (2080-2099).
Validation of the entire model cascade showed that mean annual flows, mean seasonal flows, and the mean annual flood were represented well but mean annual low flow was consistently biased. The significant changes in New Zealand freshwater catchments were found to be consistent with previous studies and will benefit water resources planning in a changing environment.
General Comments
It was a pleasure to review this outstanding publication. The introduction provided a robust literature review, the methods were appropriate to support the research goals, the results were clearly explained and supported by figures/tables, and the discussion explained the results in the context of related research and future directions.
Minor suggestions
- The Figures are reproduced each time the Figure number is referenced within the text. For example, Figure 2 appears four times.
- Line 57: a word is missing between “national importance” and “national policy”
- Line 82: It would be helpful to also summarize temperature in this paragraph
- Line 134: Add a period after New Zealand
- Figure 2: add axis labels
- Line 477: missing word(s) between “cascade” and “its”
Author Response
Response to Reviewer 1 Comments
I thank the reviewer for their constructive feedback. Point by point responses are below.
Point 1: The Figures are reproduced each time the Figure number is referenced within the text. For example, Figure 2 appears four times.
Response 1: I cannot see what the reviewer is referring to, unfortunately. In neither the .doc file submitted nor the one on the MDPI website can I see the error. Could this be an artifact of converting to PDF, perhaps? If the Editor could advise what is in error I would readily fix it.
Point 2: Line 57: a word is missing between “national importance” and “national policy”
Response 2: Deleted “national policy” and clarified the reference.
Point 3: Line 82: It would be helpful to also summarize temperature in this paragraph
Response 3: Good point. Details have been added.
Point 4: Line 134: Add a period after New Zealand
Response 4: Added.
Point 5: Figure 2: add axis labels
Response 5: Added “Modelled” and “Observed”.
Point 6: Line 477: missing word(s) between “cascade” and “its”
Response 6: Added “and”.
Reviewer 2 Report
Dear author,
Thanks a lot for your interesting manuscript.
As this research focuses on river hydrology under climate change in New Zealand, the topic of the paper seems relevant for publishing in Water. The introduction section sufficiently provides information related to climate change and its impacts on river hydrology. The description of the study area, data, and methodology applied in this manuscript are explained in details well. The results are adequate for the purpose of the study and were structured well. The discussion was expressed sufficiently. The conclusions were truly supported by the results. Hence, I recommend this manuscript for publication in Water.
Sincerely,
Author Response
I thank the reviewer for their review and I appreciate their expression of confidence in the manuscript. As no shortcomings are raised nor changes recommended, no detailed response is needed.